# Combination of Deep Cross-Stage Partial Network and Spatial Pyramid Pooling for Automatic Hand Detection

**Christine Dewi** [1,*] and **Henoch Juli Christanto** [2,*]

1   Department of Information Technology, Satya Wacana Christian University, Salatiga 50711, Indonesia
2   Department of Information System, Atma Jaya Catholic University of Indonesia, Jakarta 12930, Indonesia
*   Correspondence: christine.dewi@uksw.edu (C.D.); henoch.christanto@atmajaya.ac.id (H.J.C.)

**Abstract:** The human hand is involved in many computer vision tasks, such as hand posture estimation, hand movement identification, human activity analysis, and other similar tasks, in which hand detection is an important preprocessing step. It is still difficult to correctly recognize some hands in a cluttered environment because of the complex display variations of agile human hands and the fact that they have a wide range of motion. In this study, we provide a brief assessment of CNN-based object identification algorithms, specifically Densenet Yolo V2, Densenet Yolo V2 CSP, Densenet Yolo V2 CSP SPP, Resnet 50 Yolo V2, Resnet 50 CSP, Resnet 50 CSP SPP, Yolo V4 SPP, Yolo V4 CSP SPP, and Yolo V5. The advantages of CSP and SPP are thoroughly examined and described in detail in each algorithm. We show in our experiments that Yolo V4 CSP SPP provides the best level of precision available. The experimental results show that the CSP and SPP layers help improve the accuracy of CNN model testing performance. Our model leverages the advantages of CSP and SPP. Our proposed method Yolo V4 CSP SPP outperformed previous research results by an average of 8.88%, with an improvement from 87.6% to 96.48%.

**Keywords:** hand detection; Yolo V4; CSP; SPP; Densenet; Resnet; convolutional neural network; object detection

## 1. Introduction

The human hand is extremely crucial when individuals communicate with one another and with their surroundings in ordinary living. To recognize hand movements and human activities, the location and direction of the human hand must be carefully observed and recorded [1]. The ability to accurately identify hands in images and videos can support a wide range of visual processing tasks, including gesture and scene understanding. Unfortunately, due to the enormous variance in hands in images, it is harder to identify hands in uncontrolled environments [2]. Various orientations, shapes, and sizes are available to the hand in a very clear way. The increased variation in hand appearance is further accentuated by occlusion and motion blur [3]. In many computer vision applications, such as human–computer interaction, sign language recognition [4], and hand action analysis [5], comprehensive hand gesture recognition in cluttered environments is a critical task to accomplish.

Given the great variety in drawing hands in realistic scenarios, hand recognition is a very difficult task. This is a challenge that has not yet been resolved in the field of machine learning and artificial intelligence. Many approaches to dealing with this problem have been proposed by researchers, some of which were inspired by the recently developed deep learning approach for object detection and recognition [6,7]. However, despite significant progress, these challenges persist, even with current deep learning approaches [8,9]. Conventionally, hand region detection techniques have relied on low-level image features including skin color [10] and shape [11] to identify the hand region.

To convey meaningful information, a static hand gesture requires only one image, whereas a dynamic hand movement requires a sequence of frames to perform a single

movement. In specifically, the purpose of this effort is to overcome the challenge of detecting and recognizing static hand gestures, including postures executed with bare hands to indicate specified interpretations. Throughout this study, we attempt to address the problem of localization and recognition of static hand movements in two stages. The first stage of our proposed approach deals with the accurate location of the hand and extraction of the hand from the rendered image through the use of bounding boxes, which are generated in the second stage. It is necessary to have sufficient data for this step to build a reliable model. For this reason, we compile an extensive hand detection dataset covering hands in a variety of complex environments.

Yolo is a CNN approach that is extremely efficient and works exceptionally well for real-time object detection [12,13]. In addition to assisting with feature extraction, neural networks may also assist in understanding the significance of motion, as well as recognizing the target object [14,15]. Hand detection models combining convolutional neural networks (CNNs) with cross-stage partial (CSP) networks and spatial pyramid pooling (SPP) are summarized and examined in this paper for hand recognition. We fine-tuned it using the Oxford hand dataset, which is a well-known and advanced hand dataset. Some of the CNN models include Densenet, Yolo V2, Resnet 50, Yolo V4, and Yolo V5.

The primary objectives of this research were as follows: (1) to present a brief survey of object detection algorithms based on CNN, specifically Densenet Yolo V2, Densenet Yolo V2 CSP, Densenet Yolo V2 CSP SPP, Resnet 50 Yolo V2, Resnet 50 CSP, Resnet 50 CSP SPP, Yolo V4 SPP, Yolo V4 CSP SPP, and Yolo V5; (2) to analyze and discuss in detail the benefits of CSP and SPP in each algorithm; (3) to perform experiments showing that Yolo V4 CSP SPP produces the best accuracy through the addition of CSP and SPP layers, which can improve the performance of all models.

The remainder of this work is organized in the following manner: Section 2 provides a brief overview of some related work in the field of hand identification; Section 3 describes an overview of the proposed architecture as well as an explanation of the functions of all of its modules; Section 4 discusses and compares our experimental results with previous research results; Section 5 concludes by presenting the findings of this study, as well as recommendations for further research.

## 2. Related Work

Section 2 offers a comprehensive assessment of pertinent research in the field of hand detection and recognition techniques, including systems that rely on typically constructed features, in addition to those that employ deep learning.

### 2.1. Hand Detection and Recognition

Many hand recognition algorithms in the past depended on skin color segmentation to distinguish and extract hands from the backdrop of images, which was inefficient. In their papers [16,17], Dardas et al. suggested a thresholding technique for fragmenting hands in the hue, saturation, and value (HSV) color space after extracting other skin regions, such as the face, from the source image. Girondel and colleagues [18] experimented with a variety of color spaces and discovered that the Cb and Cr channels in the YCbCr color space performed well in the skin detection task. Sigal et al. [10] proposed the Gaussian mixture model, which performed admirably under a variety of lighting conditions. In order to precisely detect the presence of the hand, Mittal et al. [19] developed a method based on a mixture of deformable parts. According to Karlinsky et al. [20] a method for locating hands that relies on sensing the relative locations of the hand and other human body parts in order to locate them was proposed.

The most recent breakthroughs in emerging color-depth camera-based detection systems, such as the Microsoft KinectTM, have made significant contributions to hand gesture recognition, such as hand extraction utilizing depth data, and other applications. With its excellent computer vision and speech models, smart AI sensors, and a variety of strong SDKs, Microsoft's Azure Kinect provides a cutting-edge development kit for spatial com-

puting applications. Sensor SDK and Body Tracking SDK are the major SDK modules. The Body Tracking SDK is built on a complicated deep learning model that allows body segmentation, human skeleton reconstruction, human body instance detection, and real-time body tracking. A power drill chunk pose can be best estimated by looking at the n.8 (left hand) and n.15 (right hand) kinect hand joints, according to [21]. As a result of multiple testing, these joints were ruled out due of their poor orientation predictions. When it came to the actual power drill in the scenario, however, the model interpreted it as a part of the operator's hand most of the time.

On the other hand, the You Only Look Once (Yolo) framework takes a different approach to object recognition. The bounding boxes and class probabilities for these boxes are predicted for the entire image in a single instance. Yolo's tremendous speed is its most significant advantage. Yolo has a processing speed of 45 frames per second, which is impressive. Comparing Yolo to Microsoft Kinect, we may say that Yolo better understands object representations in general. Single-stage detection means that all the components of the object identification pipeline are unified into a single neural network, making Yolo appropriate for real-time detection. The Kinect sensor has limited practical range and is difficult to implement, while Yolo is easy to use and extend.

To classify images, Keskin et al. [22] employed depth images to obtain scale-invariant form characteristics, which they then input into a per-pixel random forest (RDF) classifier. In response to the recent success of convolutional neural networks (CNNs), researchers have suggested a slew of object detection and recognition methods that are based on CNNs. Utilizing CNN, it is possible to effectively address the multiscale and various rotations problems for hand identification. Le et al. [23] introduced that multiscale feature maps can be effectively combined for detection and classification to prevent small hand loss. Similarly, Qing et al. [24] presented a feature map smelting SSD that employs deconvolution to merge deep levels with shallow layers for hand identification. A compressed spatial pyramid pooling (SPP) CNN framework for recognizing gestures or finger-spelling from movies was developed by Ashiquzzaman et al. [25]. They named their model the SPP CNN model.

### 2.2. CNN for Object Detection

Compared to traditional networks, Densenet has more layers and a faster convergence rate [26,27]. Densenet should carefully consider extra functionality channels, including mono characteristics or cross-level dimensions, in order to minimize the demand for functional duplication in the network structure and increase the retrieval of features [28].

Additionally, Densenet provides several benefits. It makes it easier to reuse features and alleviates the problem of fading gradients in graphics. As a result, there are some clearly defined limitations. In the first place, each layer merely contains a combination of the feature maps that were formed by combining the procedure from the layers that occurred before it. Without taking into account the interdependencies between distinct channels [29], the operation can be carried out. Furthermore, the Densenet is built mostly of three components: the dense block, the transition layer, and the growth rate [30,31].

Resnet [32] is a deep convolutional network with residual layers whose primary idea is to skip blocks of convolutional layers by employing shortcut connections. Furthermore, Resnet is distinguished by having an extremely deep network that contains between 34 and 152 layers [33,34]. The Resnet model is characterized by the implementation of a residual network topology. Batch normalization is at the heart of ResNet's operation. The batch normalization process makes adjustments to the input layer in order to improve the overall performance of the network. The problem of covariate shift is reduced to a minimum. ResNet employs the identity connection, which aids in the prevention of the network from being affected by the disappearing gradient problem. In order to avoid overfitting, this model additionally makes use of the batch normalization technique [35].

Yolo V4 is the most recent version of the Yolo series algorithm, which was released by [36] in 2020. The Yolo V4 structure is composed of three parts: (1) the backbone

(CSPDarknet53 [37]), (2) the neck (SPP [38] and PAN [39]), and (3) the head (Yolo V3) [40]. Yolo V4 makes use of a Mish [41] activation function in the backbone of the program. Wang and colleagues [42] redesigned the Yolo V4 to obtain the Yolo V4-tiny in order to achieve the optimal speed/accuracy tradeoff. A cross-stage partial network (CSPNet) was constructed in order to attribute the problem to the presence of duplicate gradient information within the context of network optimization. Using network optimization, it is possible to reduce the complexity of the optimization while preserving accuracy [43,44]. The Focus module, the CBL module, the CSP module, the SPP module, the Concat module, and the Upsample module are the major components of the Yolo V5 network. Furthermore, there are four different network models available with Yolo V5, which are Yolo V5S, Yolo V5M, Yolo V5L, and Yolo V5X. CSP and SPP were used in conjunction with Densenet, Resnet 50, and Yolo algorithms in our research. In total, we evaluated nine distinct model combinations through the experiment.

### 2.3. Cross-Stage Partial (CSP) Network and Spatial Pyramid Pooling (SPP)

Yolo V4, an object detection system based on CSP [42], was offered as a new approach to object detection. We present a network scaling strategy that alters not only the depth, width, and resolution of the network, but also the network's structure, resulting in the construction of Scaled-Yolo V4 in this research. Yolo V4 is a real-time object detection system that runs on a general-purpose GPU. Wang et.al [42] redesigned Yolo V4 to obtain Yolo V4-CSP for the best speed/accuracy tradeoff. CSP is the Yolo-V4 backbone network used to improve the learning capacities of CNN while simultaneously reducing the computing bottleneck and memory cost. It is also lightweight, allowing it to be deployed anywhere [45,46]. CSP blocks were applied in this paper to the DarkNet53 network, and a CSP model with Densenet, Resnet, and Yolo was constructed.

SPP [38] has multiple advantages. Firstly, SPP is able to provide an adequate fixed-length output independent of the input dimensions [47,48]. Additionally, SPP makes use of multilevel spatial bins, whereas sliding window pooling only makes use of a single window size, in contrast to SPP's usage of several window sizes [49]. For the purposes of this experiment, an SPP block layer was added to the configuration files for Densenet, Resnet 50, Yolo V4, and Yolo V5. Using the same SPP block layers in the configuration file, we could also create spatial models using them. The spatial model takes advantage of downsampling in convolutional layers to get the required properties in the max-pooling layers, which are subsequently used to construct the model. Three different sizes of the max pool were applied for each image using [route]. Different layers ($-2$, $-4$, $-1$, $-3$, $-5$, and $-6$) in $conv_5$ were used in each [route]. Figure 1 illustrates the SPP architecture.

In our work, we also experimented with the latest version of Yolo V5, which is significantly different from the previous version. PyTorch is utilized in place of Darknet. This typically features the use of CSPDarknet53 as a backbone for its operations. When used in big backbones, it solves the problem of repeated gradient information and integrates gradient change into the feature map, which increases inference speed while simultaneously increasing accuracy and shrinking the model size by minimizing the number of parameters used. It makes use of the path aggregation network (PANet) as a bottleneck to increase the flow of information. Our experiment cloned the repo and install requirements.txt form github (https://github.com/ultralytics/yolov5) (accessed on 13 January 2022) in a Python $\geq$3.7.0 environment, including PyTorch $\geq$1.7. Some of the libraries based on the requirement files included matplotlib $\geq$3.2.2, numpy $\geq$1.18.5, opencv-python $\geq$4.1.1, Pillow $\geq$7.1.2, PyYAML $\geq$5.3.1, requests $\geq$2.23.0, scipy $\geq$1.4.1, torch $\geq$1.7.0, and torchvision $\geq$0.8.1.

Comparing the Yolo V5 algorithm to the Yolo V1 algorithm, the Yolo V5 approach is an improvement because it does not necessitate the usage of the entire connection layers that are required by the Yolo V1 algorithm. The depth feature of the target image is extracted using Darknet-19, which is built in C++. On the basis of this information, anchor coordinates are inserted in order to anticipate the region of interest [50].

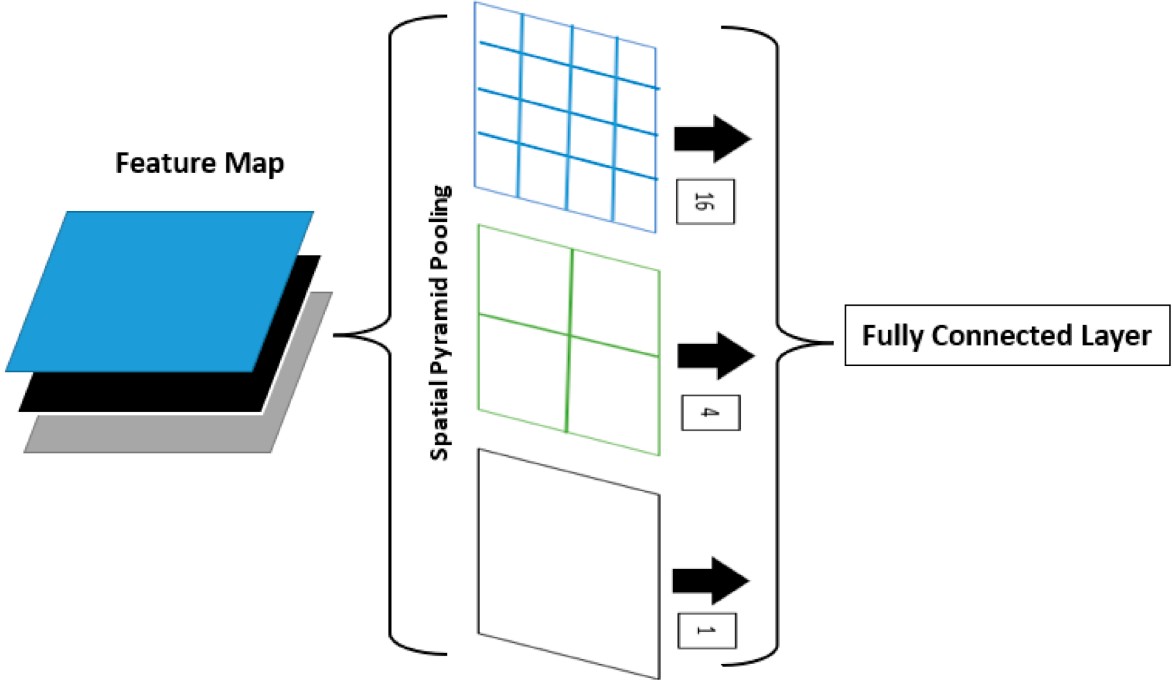

**Figure 1.** Spatial pyramid pooling (SPP) architecture.

### 3. Methodology

Our proposed approach and experimental setup are discussed in greater detail in this section. Figure 2 describes the system architecture of Yolo V4 CSP SPP. In order to create a bounding box for all hand objects, the BBox mark tool [51] was implemented. An image may have numerous marks on it because each category is labeled separately in the program. Only one detector model was utilized for detection, and each label matched to a single training model in the initial part of the experiment.

The Yolo labeling format is supported by most annotation platforms, resulting in a single annotation text file per image. Each text file contains a bounding box (BBox) annotation for each of the objects in the image, with one annotation per object in the image. They are normalized to the image size in the range of 0 to 1, depending on the size of the image. Each is represented in the following manner: *<object-class-ID> <X center> <Y center> <Box width> <Box height>*. Equations (1)–(6) serve as the foundation for the adjustment procedure.

$$dw = \frac{1}{W}, \tag{1}$$

$$x = \frac{(x_1 + x_2)}{2} \times dw, \tag{2}$$

$$dh = \frac{1}{H}, \tag{3}$$

$$y = \frac{(y_1 + y_2)}{2} \times dh, \tag{4}$$

$$w = (x_2 - x_1) \times dw, \tag{5}$$

$$h = (y_2 - y_1) \times dh, \tag{6}$$

where mage height is represented by *h*, while image absolute height is represented by *dh*; *W* denotes the image width, and *dw* denotes the image absolute width.

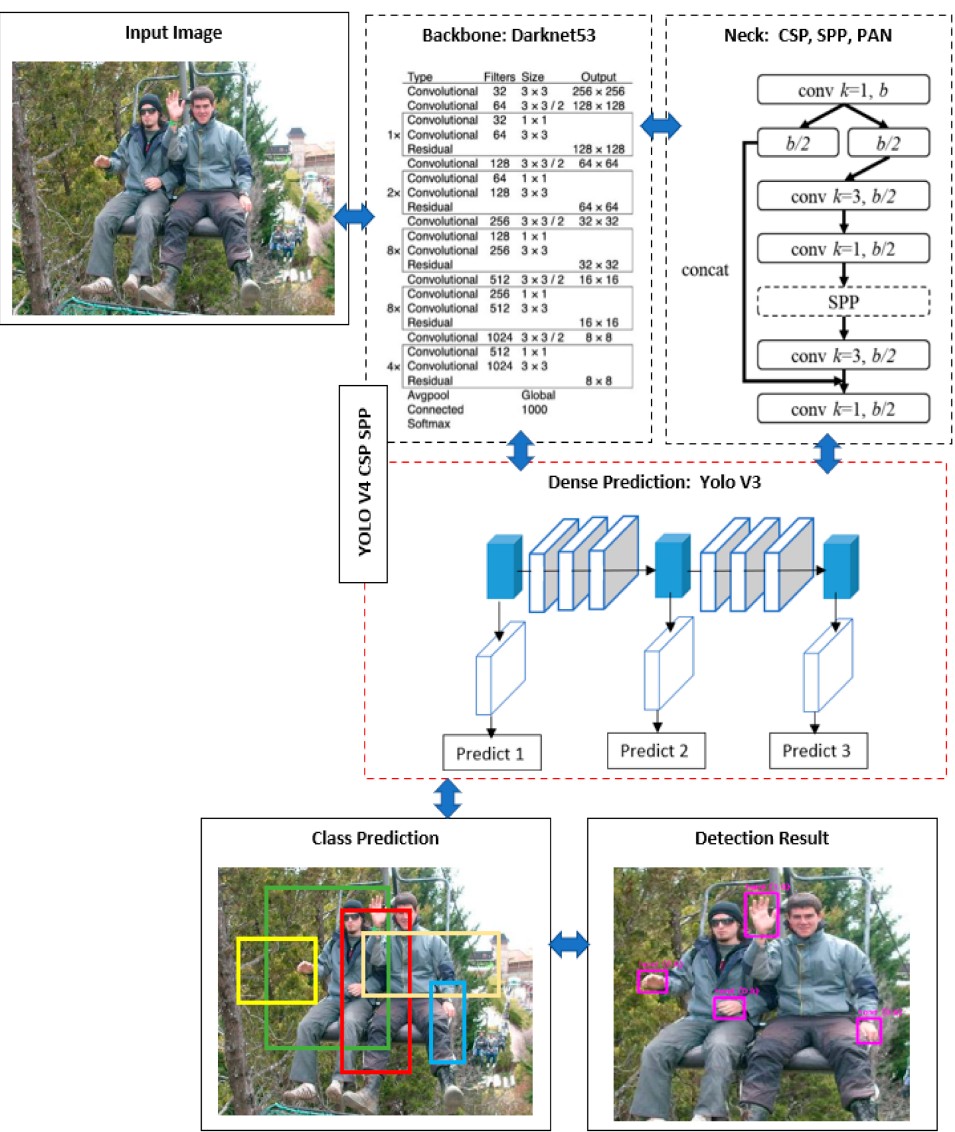

**Figure 2.** System architecture of Yolo V4 CSP SPP.

An explanation of the procedure for introducing Yolo V4 CSP SPP is provided in Algorithm 1.

---

**Algorithm 1.** Yolo V4 CSP SPP hand detection process.

---

1. Divide the input image data into (M × M) grids.
2. For each grid, create a total of K bounding boxes with an estimate of the anchor boxes in each.
3. Using CNN, extract all of the object characteristics from the image.
4. Predict the $b = \begin{bmatrix} b_x, & b_y, & b_w, b_h, & b_c \end{bmatrix}^T$ and the $class = [H1, \ H2, \ H3\ ]^T$.
5. Choose the optimum confidence $IoU_{pred}^{truth}$ of the $K$ bounding boxes with the threshold $IoU_{thres}$.
6. If $IoU_{pred}^{truth}$, it means that the bounding box includes the object. Otherwise, the bounding box does not contain the object.
7. Choose the category that has the highest estimated likelihood of being correct.
8. Non-maximum suppression (NMS) is used in conjunction with a maximum local search in order to eliminate redundant boxes and output.
9. Display the output of the object detection result.

---

Figure 3 illustrates the flow diagram of hand detection. Moreover, the hand detection process uses images from the Oxford hand dataset which includes classes H1, H2, and H3

as input data. The algorithm goes through the following phases: (1) The bounding box is used to define the boundaries of the detected targets. (2) The objects in the drawing class are linked together. In each image, the same target receives the same marks. (3) The same image results in the same consistent label for the target. (4) Non-maximum suppression (NMS) is a technique for performing a local maximum search to compress the duplicated boxes and results, and then displaying the results of the object detection process. The NMS is set up in the following way: to start, predictions are set according to their accuracy. If we examine the same class forecast and find that the IoU with the current prediction is more than 0.5, we should start with the top score and ignore the current forecast. (5) The final stage produces a categorized image labeled with the appropriate class designation.

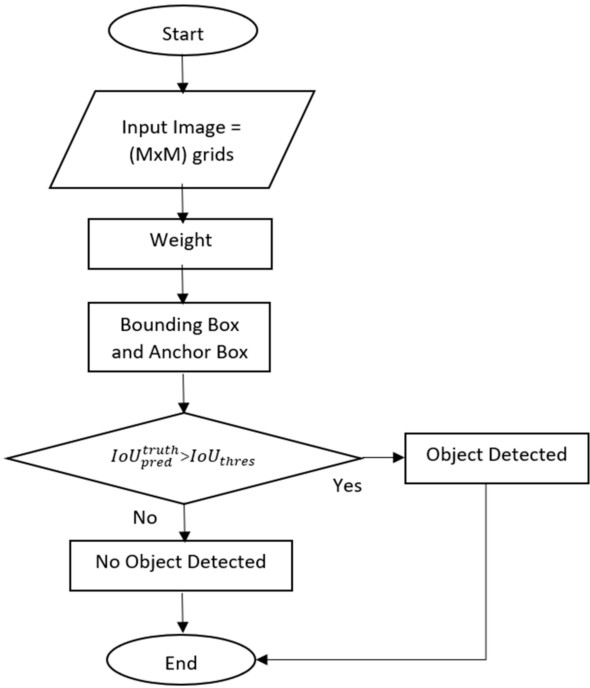

**Figure 3.** Flow diagram of hand detection.

Table 1 describes our experimental setup. Our work combined CSP and SPP with Densenet, Resnet 50, and Yolo algorithms. In total, we experimented with nine different model combinations. For Yolo V5, we simply applied the default configuration.

**Table 1.** Experimental setup.

| Model | CSP | SPP |
| --- | --- | --- |
| Densenet Yolo V2 | | |
| Densenet Yolo V2 CSP | X | |
| Densenet Yolo V2 CSP SPP | X | X |
| Resnet 50 Yolo V2 | | |
| Resnet 50 Yolo V2 CSP | X | |
| Resnet 50 Yolo V2 CSP SPP | X | X |
| Yolo V4 SPP | | X |
| Yolo V4 CSP SPP | X | X |
| Yolo V5 | X | X |

### 3.1. Oxford Hand Dataset

The Oxford hand dataset [19] is a comprehensive image dataset of hands compiled from a variety of different public image dataset sources, and it is available for free download. Each image has annotations for all of the hand instances that can be viewed clearly by

humans in that image. There are 13,050 hand instances in total over the entire dataset. Each hand instance in the training set has 11,019 data points, whereas the testing set contains 2031 data points per hand instance. During the data collection process, no restrictions were placed on the subject's attitude or visibility, nor were any restrictions placed on the surrounding environment. Each image contains annotations for all the hands that can be clearly seen by humans in the image. The annotations need to be oriented with respect to the wrist, but the bounding rectangles do not have to be aligned along any axes. A standard MATLAB "*. mat*" format is used to store the annotations for the four end points of the hand bounding box, which are represented by the files in the "annotations" folder. Boxes make up the structure, with hand-boxes representing the various indices of the cell array. We preprocess the data in this dataset to obtain the Yolo format. The dataset is divided into two parts: 70% for training and 30% for testing; it contains images of a variety of different hand objects.

Figure 4 shows a sample image of Oxford hand dataset. Furthermore, Figure 5 describes the label and correlogram of the Oxford hand dataset. Yolo V5 generates correlogram plots to display the features of the supplied image and bounding box. As we can see in Figure 5a, our experiment only has one class: "hand". Furthermore, the hand images contain nearly 6000 instances, with $x$, $y$ varying from 0.0 to 1.0, and height, weight varying from 0.0 to 0.8, as can be seen in Figure 5b.

### 3.2. Training Result

Figure 6 describes the training performance of all models in our work. The training process involving the Densenet Yolo V2 CSP SPP model is depicted in Figure 6a. A network size width of 416 and height of 416 are specified in the training configuration file. The training is processed for 9000 batches, with the max batches set to 45,000, and the policy set to steps. The weight is updated after 1000 iterations, and the learning speed is controlled by a hyperparameter that determines how much the model should change in response to the estimated error. The training process remained stable with a final average loss of 0.1337. Figure 6b shows the persistence of the training process using Resnet Yolo V2 CSP SPP. The training stage remained stable after 7000 epochs. This model used max_batches = 45,000 and mask_scale = 1, with the training loss value reaching 0.1256. In addition, Yolo V4 CSP SPP used width = 512, height = 512, channels = 3, momentum = 0.949, decay = 0.0005, angle = 0, max_batches = 6000, steps = 4800, and 5400 iterations, fluctuating with a loss value of 1.3146 (Figure 6c).

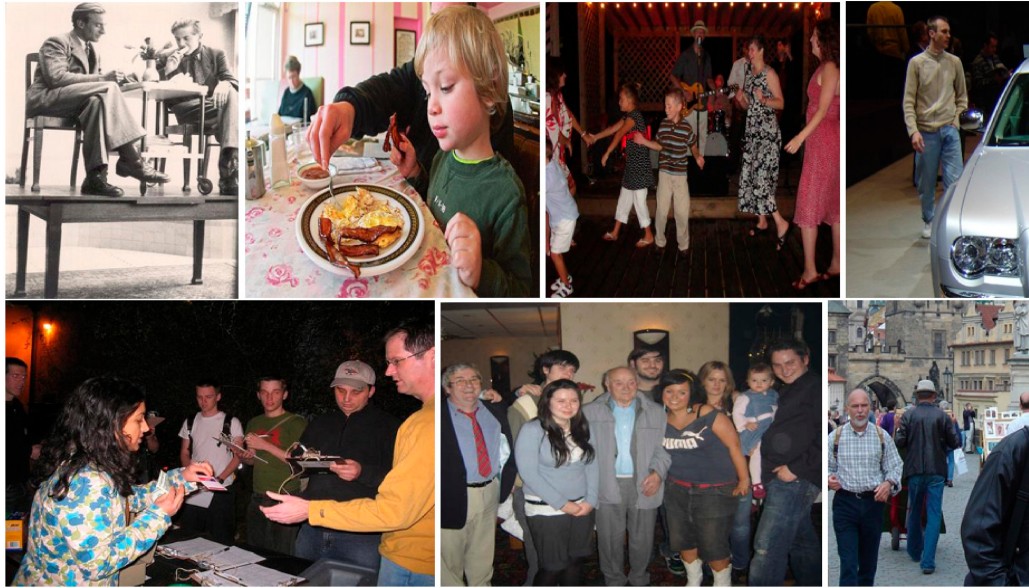

**Figure 4.** Oxford hand dataset sample images.

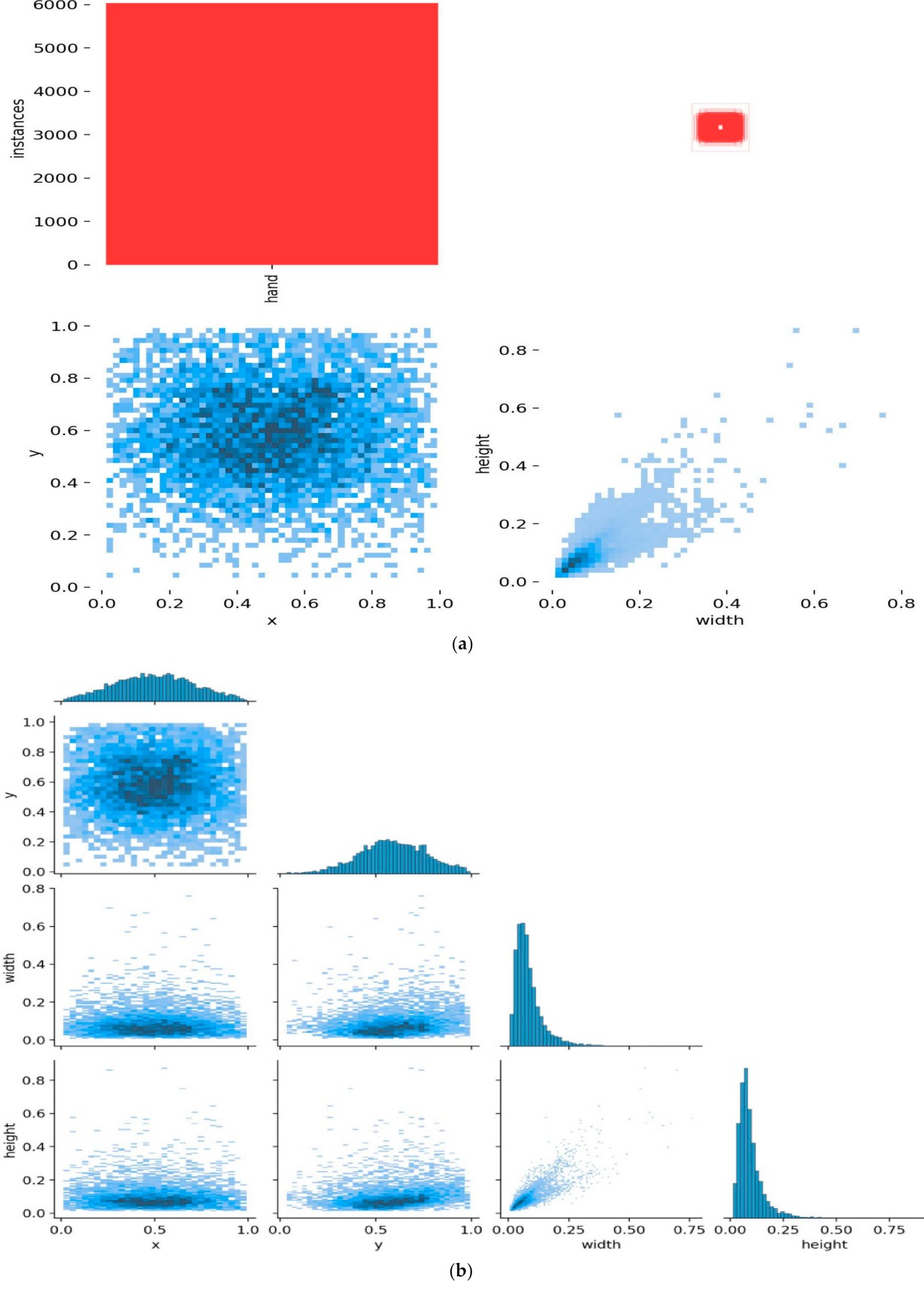

**Figure 5.** Oxford hand dataset: (**a**) labels and (**b**) correlogram.

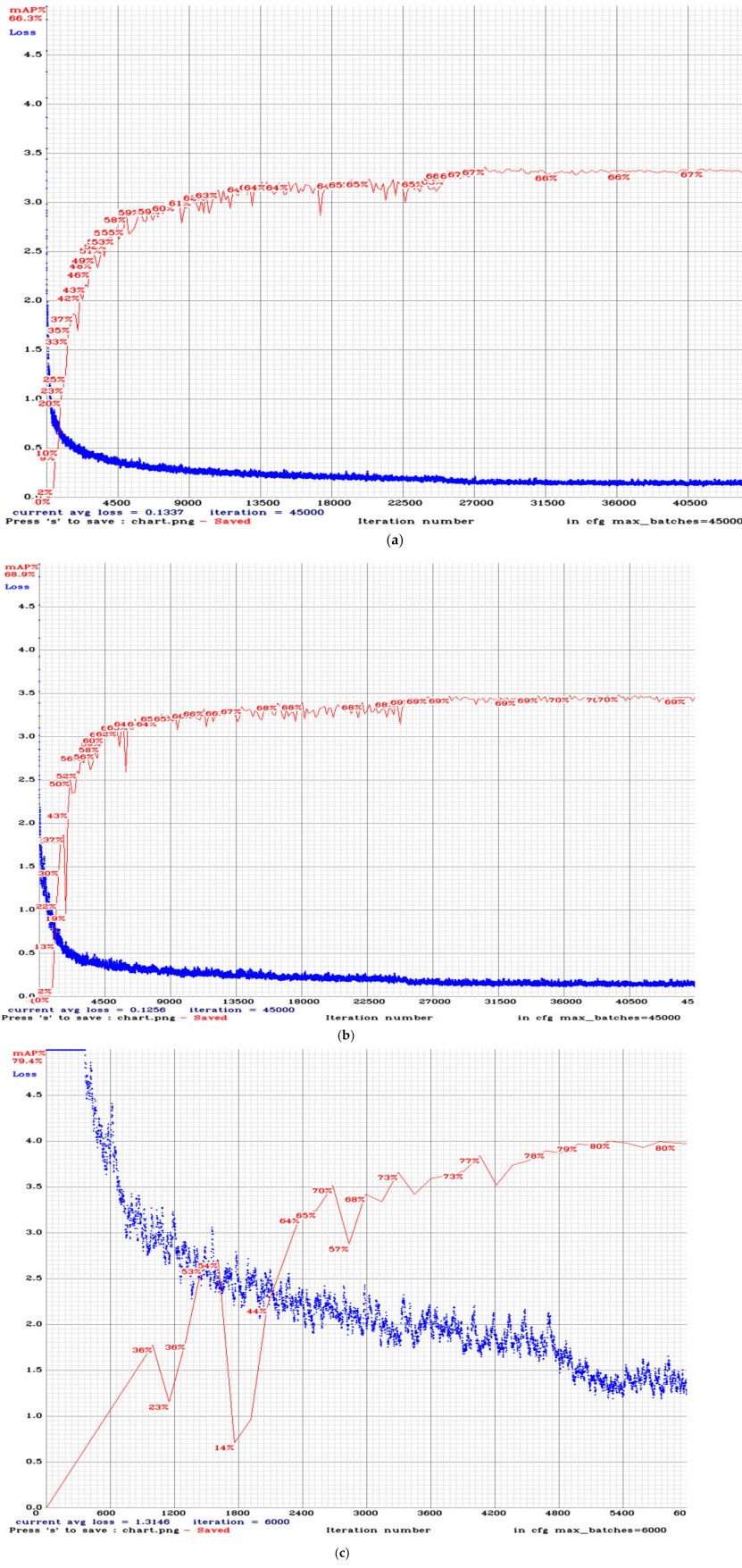

**Figure 6.** Training performance: (**a**) Densenet Yolo V2 CSP SPP; (**b**) Resnet Yolo V2 CSP SPP; (**c**) Yolo V4 CSP SPP.

Moreover, Figure 7 exhibits the training and validation performance with Yolo V5. In our experiment, we trained Yolo V5 with 30 epochs that were completed in 6615 h. Some hyperparameters in the Yolo V5 configuration file were as follows: iou_t = 0.2, anchor_t = 4.0, fl_gamma = 0.0, hsv_h = 0.015, hsv_s = 0.7, hsv_v = 0.4, degrees = 0.0, translate = 0.1, scale = 0.5, shear = 0.0, perspective = 0.0, flipud = 0.0, fliplr = 0.5, mosaic = 1.0, mixup = 0.0, copy_paste = 0.0, lr0 = 0.01, lrf = 0.01, momentum = 0.937, weight_decay = 0.0005, warmup_epochs = 3.0, warmup_momentum = 0.8, warmup_bias_lr = 0.1, box = 0.05, cls = 0.5, cls_pw = 1.0, obj = 1.0, and obj_pw = 1.0. The Yolo loss function is expressed below [52,53].

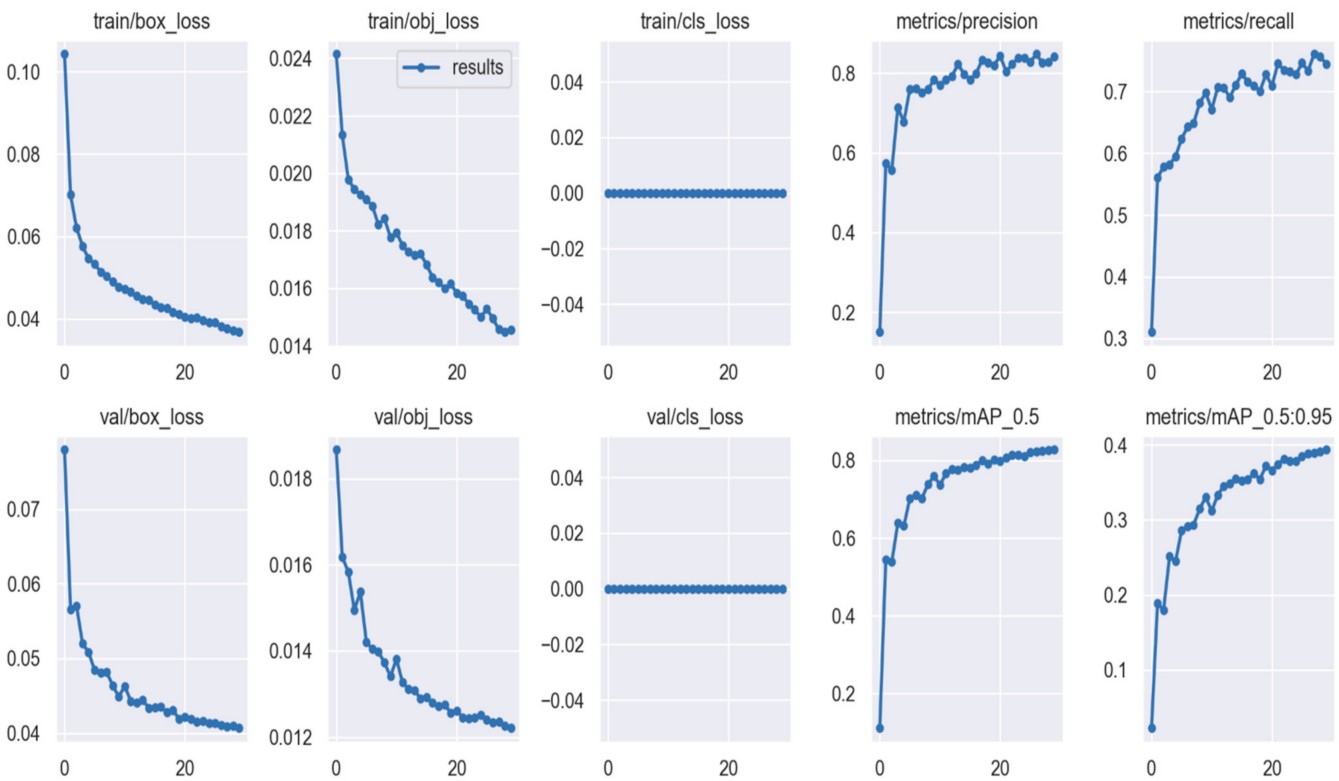

**Figure 7.** Training and validation performance with Yolo V5.

$$Yoloss = \sum_{i-0}^{s^2} coordError + iouError + classError, \tag{7}$$

$$BC(a, \hat{a}) = -[a \log \hat{a} + (1-a) \log(1-\hat{a})], \tag{8}$$

$$ST(w, h) = 2 - w \times h, \tag{9}$$

$$iouError = \sum_{i-0}^{s^2} \sum_{j-0}^{B} I_{ij}^{obj}[BC(c_i, \hat{c}_i)] + \lambda_{noobj} \sum_{i-0}^{s^2} \sum_{j-0}^{B} \{I_{ij}^{noobj}[BC(c_i, \hat{c}_i)], \tag{10}$$

$$classError = \sum_{i-0}^{s^2} I_{ij}^{obj} \sum_{c \epsilon classes} BC(p_i(c), \hat{p}_i(c)), \tag{11}$$

$$\begin{aligned} coordError = & \sum_{i-0}^{s^2} \sum_{j-0}^{B} \{I_{ij}^{obj} \times ST(w_{ij}, h_{ij}) \times [BC(x_i, \hat{x}_i) + BC(y_i, \hat{y}_i)] \\ & + \lambda_{coord} \sum_{i-0}^{s^2} \sum_{j-0}^{B} \{I_{ij}^{obj} \times ST(w_{ij}, h_{ij}) \times [(w_i, \hat{w}_i)^2 + BC(h_i, \hat{h}_i)]\}, \end{aligned} \tag{12}$$

where $(\hat{x}, \hat{y}, \hat{w}, \hat{h}, \hat{c}, \hat{p})$ are the central coordinates, width, height, confidence, and category probability of the predicted bounding box, and symbols without the cusp are real

labels. *B* symbolizes that any grid is divided into *B* bounding boxes, $I_{ij}^{noobj}$ represents that the object drops within the *j*-th bounding box of the *i*-th grids, and $I_{ij}^{noobj}$ exhibits that there are no targets in the bounding box.

In addition, *IouError* is the *IoU* problem that was encountered. The weights of the grids that contain the object and the grids that do not contain the object are different. As a result, the parameter $\lambda noobj = 0.5$ is included to reduce the influence of a high number of grids without objects on the loss estimate. *ClassError* is the classification error that occurred. Cross-entropy is used to calculate losses, and it is only effective on a grid that contains a target. Furthermore, the coordinate mistake is denoted by *CoordError*.

In order to compute the coordinates in the core point, the cross-entropy loss is applied, and the variance loss is applied for the width and height parameters. Our experiment used a value of 0.5 for $\lambda coord$, which means that the errors in width and height in the calculation were less significant. When the grid predicts the presence of an object, the computation for a coordinate error is performed [54].

Precision (P) and recall (R) [55] are represented in Equations (13) and (14) [56,57]. An outcome that is referred to as a true positive (TP) outcome occurs when the model correctly predicts the positive class. When the model accurately predicts the negative class, this is referred to as a true negative (TN). A true positive (TP) is an outcome in which the model properly predicts the positive class, but the outcome is false positive (FP). FN is an outcome in which the model forecasts the negative class erroneously, resulting in a false negative. Another evaluation index, F1 [58–60], is shown in Equation (15). The integral over the precision *p(o)* is the average mean average precision (*mAP*), as shown in Equation (16). The mean average precision (*mAP*) of an object detection model is a metric used to evaluate the model.

$$P = \frac{TP}{TP + FP}, \tag{13}$$

$$R = \frac{TP}{TP + FN}, \tag{14}$$

$$F1 = \frac{2 \times Precision \times Recall}{Precision + Recall}, \tag{15}$$

$$mAP = \int_0^1 p(o)do, \tag{16}$$

where *p(o)* is the precision of the object detection. A well-known statistic for measuring how much overlap occurs among multiple bounding boxes or masks is the intersection over union (*IoU*). *IoU* computes the overlap ratio between the boundary box of the prediction (*pred*) and ground truth (*gt*) [61].

$$IoU = \frac{Area_{pred} \cap Area_{gt}}{Area_{pred} \cup Area_{gt}}. \tag{17}$$

Table 2 shows the training performance of all models in the experiment. Yolo V4 CSP SPP obtained the maximum *mAP*, around 82.13%, with an *IoU* of 61.36%, followed by Yolo V5 at 82.7% with an *IoU* of 64.85%, and Yolo V4 at 81.5% with an IoU of 62.11%. According to Table 2, it can be concluded that the CSP and SPP layers could improve the results of Densenet, Resnet, and YoloV4 training. Our model could leverage CSP and SPP to improve *mAP* during the training process. For example, Yolo V4 achieved 81.5% *mAP* with only the SPP layer; when we combined the CSP and SPP layers, Yolo V4 CSP SPP achieved 82.13% *mAP*. The Densenet model achieved 65.65% *mAP*, which increased to 66.82% when the CSP layer was included. Furthermore, by integrating the CSP and SPP layers, the accuracy of the Densenet model was increased to 67.19%.

**Table 2.** Performance of all models during training according to precision, recall, F1-score, *IoU*, and *mAP*.

| Model | Precision | Recall | F1-Score | *IoU* (%) | *mAP*@0.50 (%) |
|---|---|---|---|---|---|
| Densenet Yolo V2 | 0.68 | 0.68 | 0.68 | 49.16 | 65.65 |
| Densenet Yolo V2 CSP | 0.71 | 0.68 | 0.69 | 50.98 | 66.82 |
| Densenet Yolo V2 CSP SPP | 0.7 | 0.68 | 0.69 | 49.96 | 67.19 |
| Resnet 50 Yolo V2 | 0.68 | 0.57 | 0.62 | 46.62 | 54.4 |
| Resnet 50 Yolo V2 CSP | 0.68 | 0.58 | 0.63 | 47.22 | 56.51 |
| Resnet 50 Yolo V2 CSP SPP | 0.67 | 0.63 | 0.65 | 47.19 | 60.99 |
| Yolo V4 | 0.83 | 0.78 | 0.81 | 62.11 | 81.5 |
| Yolo V4 CSP SPP | 0.82 | 0.76 | 0.79 | 61.36 | 82.13 |
| Yolo V4 Tiny | 0.82 | 0.46 | 0.59 | 59.2 | 56.71 |
| Yolo V5 | 0.84 | 0.74 | 0.79 | 64.85 | 82.7 |

Similarly, the Resnet 50 Yolo V2 model achieved 54.4% *mAP* with the base layer. When adding a CSP layer, the *mAP* increased to 56.51%. The accuracy of the Resnet 50 model was increased to 60.99% as a result of the integration of the CSP and SPP layers. Yolo V5 makes extensive use of cross-stage partial network (CSP) layers, which serve as the foundation for extracting highly useful characteristics from an input image. In our experiment, Yolo V5 obtained 82.7% *mAP* with an *IoU* of 64.85%.

## 4. Results and Discussion

Table 3 presents information about the testing accuracy using the Oxford hand dataset during the experiments. Overall, Yolo V4 CSP SPP was more accurate than the other CNN models. Yolo V4 CSP SPP had the highest average accuracy at 96.48%, followed by Yolo V4 with 92.99% average accuracy, and Densenet Yolo V2 CSP SPP with 91.27% average accuracy. Furthermore, in the Resnet 50 model series, Resnet 50 Yolo V2 CSP SPP achieved the highest average accuracy of 86.53%. According to the experimental results in Table 3, it can be concluded that the CSP and SPP layers increased the accuracy of the test performance of all CNN models. In addition, Yolo V4 Tiny achieved the minimum average precision accuracy of 70.17%.

**Table 3.** Testing performance using Oxford hand dataset.

| Model | Precision | Recall | F1-Score | *IoU* (%) | AP@0.50 (%) |
|---|---|---|---|---|---|
| Densenet Yolo V2 | 0.9 | 0.91 | 0.91 | 70.91 | 90.19 |
| Densenet Yolo V2 CSP | 0.9 | 0.9 | 0.9 | 69.44 | 90.45 |
| Densenet Yolo V2 CSP SPP | 0.91 | 0.91 | 0.91 | 70.86 | 91.27 |
| Resnet 50 Yolo V2 | 0.84 | 0.82 | 0.83 | 59.96 | 82.25 |
| Resnet 50 Yolo V2 CSP | 0.83 | 0.82 | 0.83 | 59.64 | 83.01 |
| Resnet 50 Yolo V2 CSP SPP | 0.84 | 0.86 | 0.85 | 62.49 | 86.53 |
| Yolo V4 | 0.9 | 0.92 | 0.91 | 69.8 | 92.99 |
| **Yolo V4 CSP SPP** | **0.92** | **0.95** | **0.94** | **72.34** | **96.48** |
| Yolo V4 Tiny | 0.89 | 0.57 | 0.7 | 66.14 | 70.17 |
| Yolo V5 | 0.83 | 0.77 | 0.80 | 66.51 | 84.4 |

Figure 8 shows the detection result using the Oxford hand dataset when employing the Yolo V4 CSP SPP. Our proposed method could recognize all hands accurately. The CSP network is meant to attribute the problem to the redundant gradient information within network optimization, allowing for a significant reduction in complexity while preserving accuracy. Pooling pyramids are also resistant to deformation caused by objects. Because of these advantages, SPP-net should be able to improve the performance of all CNN-based image classification algorithms in general. Our model utilized CSP and SPP, which allowed all tested models to obtain the highest accuracy results. Even though our model was capable of accurately detecting hands, there were still occasional false positives

and missing detections. For example, in Figure 9, our proposed method could not detect all hand objects in the image because of some reasons: (1) when two hands were near one another, they could be mistakenly identified as a single hand; (2) in situations where the human hand was substantially obscured, the scale was too small, or the appearance was too dark, it is possible that detection was missed.

Average precision (AP) was used to assess overall performance. The overlap of the hand detection with the basic truth bounding box determines whether the detection is real or false. If the overlapping score of a box is greater than 0.5, the box is considered positive. When the bounding rectangle around the ground truth and hand bounding box is not parallel to the axis, the overlap ratio between the bounding rectangles is calculated. In our work, the proposed approach was compared to approaches described in [1,2,19,23,61,62]. We emploed the evaluation metric described in [62], which uses the normal average precision when the threshold of *IoU* is 0.5. (AP50). We compared the findings of our methodology with those obtained using state-of-the-art methodologies on the Oxford hand dataset. Table 4 shows the AP50 of the state-of-the-art approaches on the Oxford hand dataset. Our proposed approach, Yolo V4 CSP SPP, achieved an average precision of 96.48% when the threshold of IoU was set to 0.5. This was the highest average precision compared to other previous research result. Our proposed method Yolo V4 CSP SPP increased the average precision compared to the state-of-the art method proposed in [1] by 8.88% (from 87.6% to 96.48%).

The BFLOPS, workspace size, and layer for each CNN model in this study are illustrated in Figure 10, providing a comparison of the three variables. Yolo V4 CSP SPP achieved an optimal BFLOPS of 76.2, whereas Yolo V4 Tiny presented the minimum of 6878 BFLOPS (Figure 10a). The Densenet model required the largest workspace size of 104.86 MB (Figure 10b). Furthermore, Yolo V5 required the smallest workspace size of ~14.3 MB, Resnet 50 required 26.22 MB, and Yolo V4 required 52.43 MB. The combination of CSP and SPP layers increased the size of te original model. Densenet Yolo V2 CSP SPP contained 321 layers, Resnet 50 Yolo V2 CSP SPP contained 84 layers, and Yolo V4 CSP SPP contained 176 layers according to the weight file (Figure 10c).

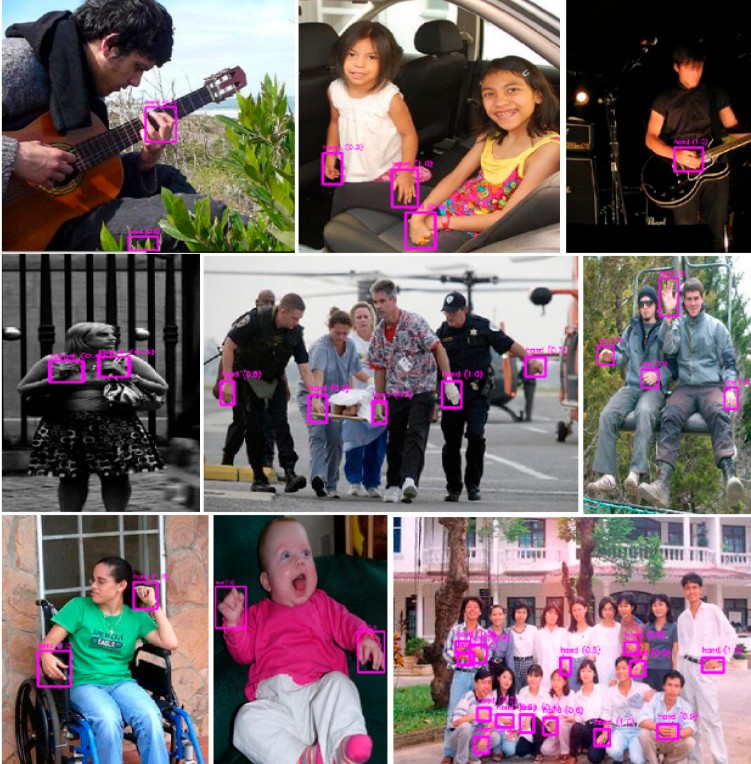

**Figure 8.** The detection results of Yolo V4 CSP SPP on the Oxford hand dataset.

**Table 4.** Comparison of the performance of various methods for Oxford hand dataset detection.

| Author | Model | AP@0.50 (%) |
|---|---|---|
| Mittal et al. [19] | Multi Proposal with NMS | 48.2 |
| Deng et al. [62] | RPN and Rotation Estimation | 58.1 |
| Le et al. [23] | MS-RFCN | 75.1 |
| Narasimhaswamy et al. [2] | Hand-CNN | 78.8 |
| Yang et al. [61] | SSD Hand | 83.2 |
| Xu et al. [1] | Hybrid Detection with GAN | 87.6 |
| Our method | Yolo V4 CSP SPP | 96.48 |

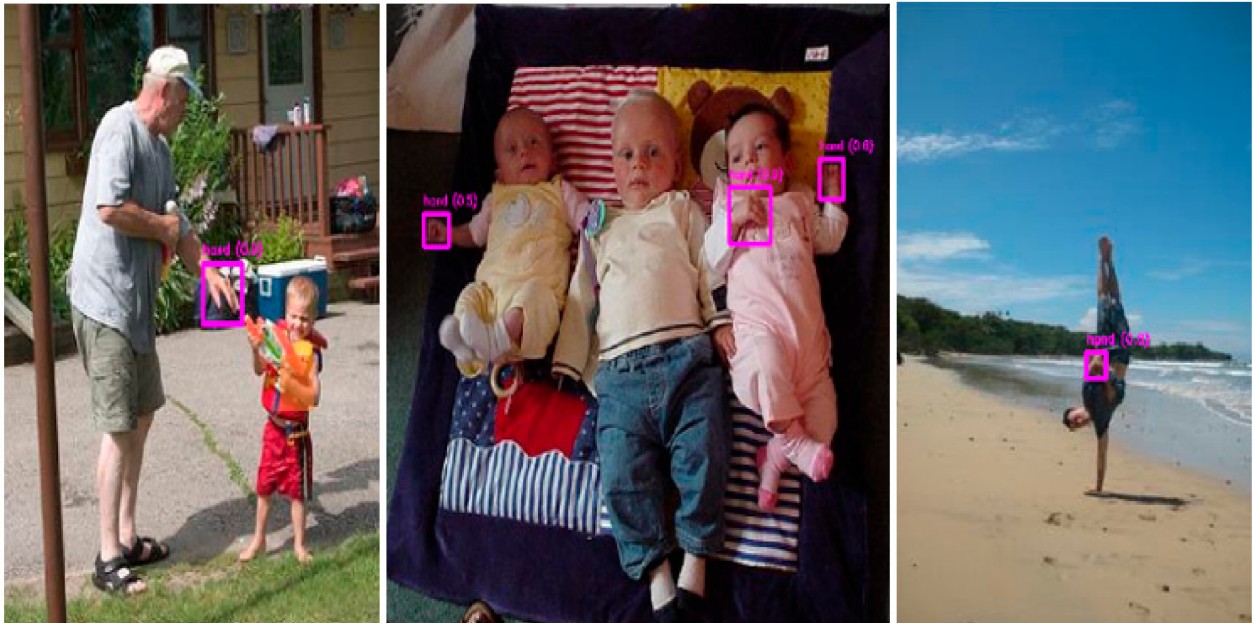

**Figure 9.** Yolo V4 CSP SPP failure cases on Oxford hand dataset.

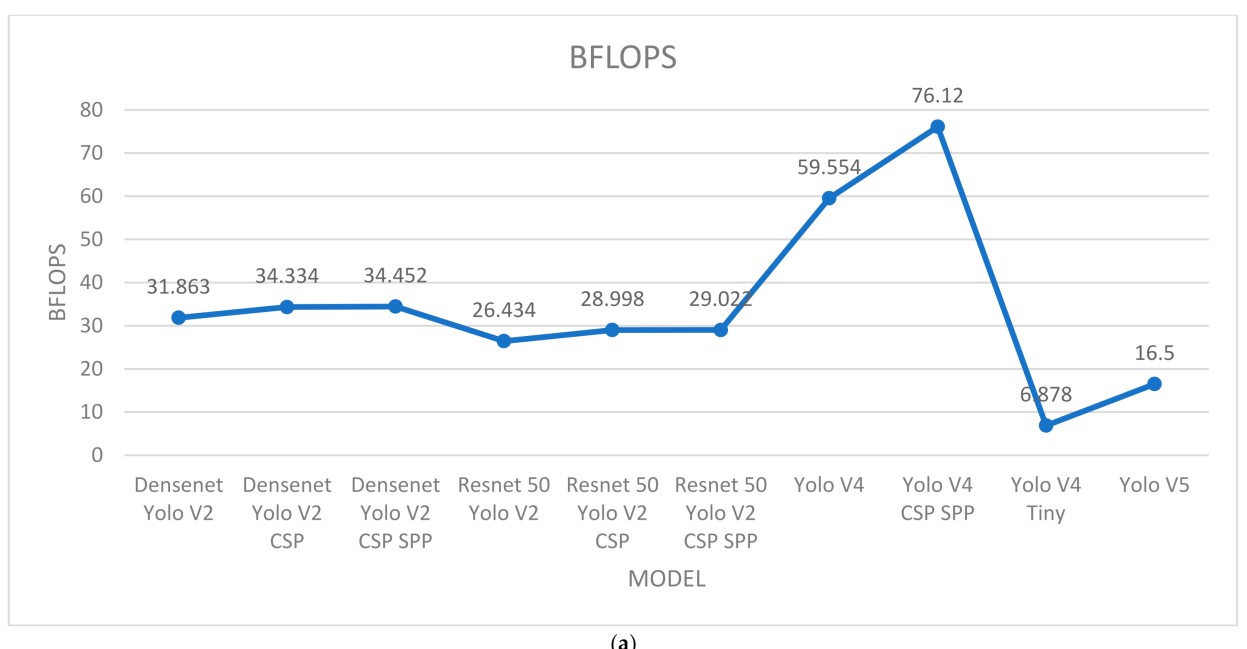

(a)

**Figure 10.** *Cont.*

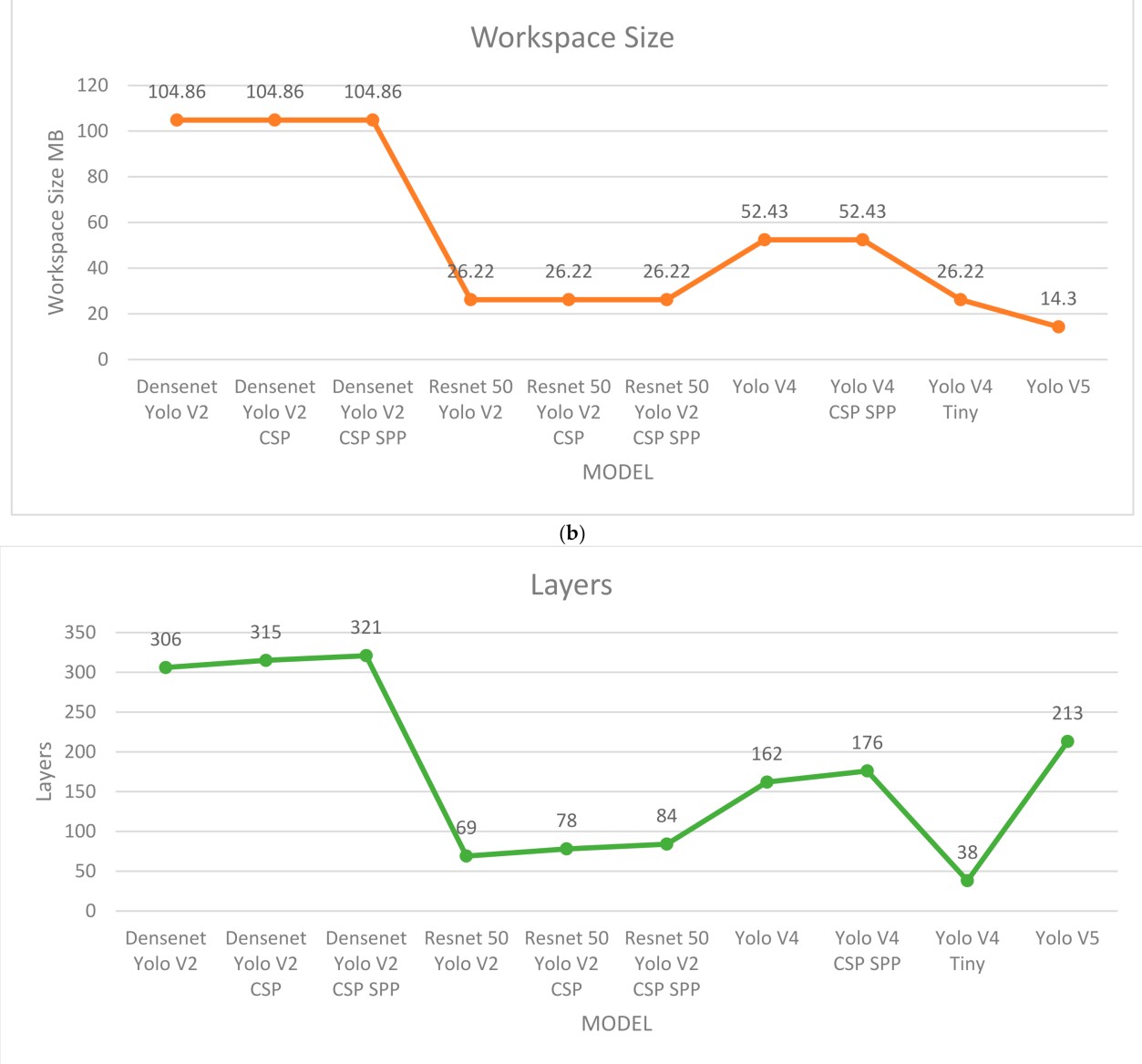

**Figure 10.** Comparison of (**a**) BFLOPS, (**b**) workspace size, and (**c**) number of layers.

## 5. Conclusions

This paper discussed a brief examination of CNN-based object identification algorithms, specifically Densenet Yolo V2, Densenet Yolo V2 CSP, Densenet Yolo V2 CSP SPP, Resnet 50 Yolo V2, Resnet 50 CSP, Resnet 50 CSP SPP, Yolo V4 SPP, Yolo V4 CSP SPP, and Yolo V5. In each algorithm, we examined and described in detail the advantages of CSP and SPP. Our experiments showed that Yolo V4 CSP SPP provided the highest level of precision. The experimental results indicated that the CSP and SPP layers helped to increase the accuracy of the CNN models' test performance. Our model leveraged the advantages of CSP and SPP. Compared to the proposed advanced approach in [1], our proposed method Yolo V4 CSP SPP improved the average precision by 8.88% (from 87.6% to 96.48%). Yolo V4 CSP SPP achieved an optimal BFLOPS of 76.2 with a load of 176 layers from the weight file.

In the Yolo framework, object recognition is handled differently. In each prediction, the bounding box of the image and the class probabilities are predicted. Yolo's greatest asset is its incredible speed. An astonishing 45 frames per second can be achieved by Yolo. We can say that Yolo achieves object representation better than other approaches. All components

of the item identification pipeline are combined into a single neural network, which makes Yolo suitable for real-time detection, easy to use, and expandable.

However, our proposed method cannot differentiate between right and left hands. Therefore, it is very important to address this issue in the future. Furthermore, we will combine hand detection with explainable artificial intelligence in future research.

**Author Contributions:** Conceptualization, C.D.; and H.J.C.; formal analysis, C.D.; and H.J.C.; investigation, C.D.; methodology, C.D.; project administration, C.D.; and H.J.C.; software, C.D.; validation, C.D.; visualization, C.D.; writing—original draft, C.D.; writing—review and editing, C.D.; and H.J.C. All authors have read and agreed to the published version of the manuscript.

**Funding:** This research received no external funding.

**Institutional Review Board Statement:** Ethical review and approval were waived for this study, due to reason that we use the public and free Oxford Hand Dataset.

**Informed Consent Statement:** Written informed consent was waived for this study due to reason that we use the public and free Oxford Hand Dataset.

**Data Availability Statement:** Oxford Hand Dataset (https://www.robots.ox.ac.uk/~vgg/data/hands/ (accessed on 9 July 2022) and https://drive.google.com/drive/folders/11zS5IYJAdyKrYD127-RPLNaeJgK1eRaP?usp=sharing, (accessed on 9 July 2022)).

**Acknowledgments:** The authors would like to acknowledge all the colleagues and partners from Satya Wacana Christian University, Atma Jaya Catholic University as well as others that took part in this work.

**Conflicts of Interest:** The authors declare no conflict of interest.

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
