# Peer review of "Combination of Deep Cross-Stage Partial Network and Spatial Pyramid Pooling for Automatic Hand Detection"

_2504-2289, doi:10.3390/bdcc6030085_

Round 1

Reviewer 1 Report

Thank you for inviting me to be a reviewer of the manuscript entitled Combination of Deep Cross Stage Partial Network and Spatial Pyramid Pooling for Automatic Hand Detection. This document is really impressive in terms of your efforts to demonstrate the power of your study.

The study focuses on the introduction and comparison of algorithms for the detection of objects, specifically the human hand.

The introductory chapters of this study present a theoretical basis based on a Literary review. Section 2 offers a comprehensive assessment of pertinent research in the field of hand detection and recognition techniques, including systems that rely on typically constructed features in addition to those that employ deep learning. Chapter 3 describes the proposed approach and experimental setup. This is followed by a section devoted to a discussion of the results and an overview of the study. The presented study contains a large number of clear graphs, tables and diagrams that clearly present the abortion results and their differences.

The presented diagrams and other graphics present the study well. For example, the schematic in Figure 2. Yolo V4 CSP SPP system architecture. It is clear and concise and important for describing the system architecture.

In the study, I see great potential for further follow-up research.

However, some passages of the study are very descriptive and lengthy. This is sometimes confusing. Therefore, I would suggest shortening and simplifying them.

This study refers to 61 scientific references, resources and publications. The references used are current and of sufficient quality, and are a suitable tereotic basis for this study.

This study represents a contribution in this area of research.

The basic ideas of the submitted manuscript are fascinating and interesting.

Reviewer 2 Report

Dear authors,

The reading of this manuscript was quite enjoyable for me. The paper provides a brief assessment of CNN-based object identification algorithms, specifically Densenet Yolo V2, Densenet Yolo V2 CSP, Densenet Yolo V2 CSP SPP, Resnet 50 Yolo V2, Resnet 50 CSP, Resnet 50 CSP SPP, Yolo V4 SPP, Yolo V4 CSP SPP, and Yolo V5.

The authors examined the advantages of CSP and SPP and described in detail each algorithm.

The following aspects must be improved and revised:

1. The experimental methodology is described with sufficient data. Maybe the research should be completed with the representation of the algorithms in the form of a logic diagram.

2. In the section 3.1 Oxford Hand Dataset it is necessary to provide an interpretation of the figure 4 (a)and(b).

3. The conclusions are briefly presented and do not present recommendations for use in practice.

In general, this work has contributed to and added value to the current body of knowledge.

Reviewer 3 Report

The manuscript was prepared correctly. Only minor additions are required.

Detailed comments:

- Microsoft Kinect (line 98) is given as an example of a hardware-based implementation of user gesture detection. Gestures can also be recognized without any image processing. Application of systems such as MPU-9150, LSM303D, ADXL345 or ADXL193 can also solve this problem. It would be sufficient to use the Hidden Markov model, dynamic time warping (DTW) or finite state machine (FSM). How can the author evaluate and compare her solution and other approaches for use in battery-powered devices? A comment on this would be required.

- Details on the use of PyTorch must be provided (line 177). At least what package was used. Was PyTorch used locally or were cloud platforms used?

- The description to Figure 4 must be extended.

- The conclusion in line 326 requires a broader justification.

- The conclusions chapter needs to be expanded. Is practical implementation envisaged?
